

# Catalogue of phonon modes in several cuprate high-temperature superconductors from density functional theory

**Noah J. Jabusch, Pavan Dayal and Alexander F. Kemper**⋆

Department of Physics, North Carolina State University, Raleigh, North Carolina 27695, USA

⋆ akemper@ncsu.edu

## Abstract

Cuprates are promising candidates for study in developing higher temperature super-conductors. A thorough understanding of a material's phonon modes enables further investigation of its emergent properties, however, no complete reference of the phonon modes exists. Here, using density functional theory, we evaluate the phonon frequencies and atomic displacements for $La_2CuO_4$, $Bi_2Sr_2CuO_6$, and $Bi_2Sr_2CaCu_2O_8$ in their tetragonal structures. The phonon modes for all materials agree with those expected from space group symmetry and display instabilities corresponding to known low- temperature structural phase transitions.

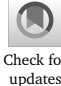

# 1 Introduction

The high-$T_c$ cuprates remain a heavily studied class of materials several decades after their discovery. They host a variety of complex phases aside from superconductivity, including a pseudogap phase, and charge/spin density waves [1]. From a crystallographic perspective, the materials are varied and complex, with large unit cells [2]. The structures host a concomitantly large number of phonon modes, several of which interplay with the various phases [1]. Their frequencies shift as orders appear, as do their relative strengths in various spectra. With the advent of resonant inelastic X-Ray scattering more detailed aspects of phonons and electron-phonon coupling in these materials are revealed [3–5]. Finally, the recently developed time-resolved experiments have further put phonons into the spotlight as the heat sink whose coupling to the electrons determines the time domain dynamics [6–9], and as a potential driver of light-induced superconductivity [10–12].

The role of electron-phonon interactions in cuprate superconductivity remains a matter of intense discussion. Early doubt regarding its relevance has been supplanted by further investigation into how it may impede or enhance superconductivity [13–15]. Density functional theory (DFT) calculations tend to find fairly small coupling constants [13, 15], which has been supported by time-dependent photoemission spectroscopy as well [9]. However, tunneling spectroscopy measurements to compute the Eliashberg function $\alpha^2 F(\omega)$ produce a substantially larger value [16–18], which may be attributed to an underestimation by DFT-based methods due to the lack of strong electron-electron interactions in these methods [16].

Of particular interest are phonons involving the Cu-O plane, where superconductivity in the cuprates is thought to occur, as these modes contribute the majority of the interaction strength [9, 13]. This aligns with several angle-resolved photoemission spectroscopy (ARPES) experiments, which have attributed kinks in the spectra to phonon interactions, specifically involving the in-plane oxygen and copper atoms [14, 19–23]. Several studies have noted that in highly correlated materials such as the cuprates, even moderate coupling can contribute to polaron formation [15, 24–27]. In doing so, these phonons can mediate d-wave pairing, although the combination of these effects may impede superconductivity [24, 26, 28].

Phonons have also been identified as a contributing factor in investigations of charge density wave (CDW) order in several cuprates, as several modes soften near the wave vector of the CDW [22, 29–31]. Ref. [31] specifically investigates softening of the buckling mode, which suggests the formation of a uniaxial CDW in $YBa_2Cu_3O_7$ (YBCO). The CDW order competes strongly with the superconducting order, strengthening under high magnetic fields which weaken or destroy superconductivity [29, 32]. Interactions with charge density provide another complication in unraveling the effect of Cu-O plane phonons on these materials.

Notwithstanding the significant attention that these materials have received, there is as yet no comprehensive reference for the phonon modes in these materials; rather, the information is spread across a wide array of papers. This paper partially addresses this gap in knowledge, following a similar approach to Ref. [13] which studies YBCO. Using density functional theory (DFT), we investigate the phonons in three parent compound representatives: $Bi_2Sr_2CuO_6$ (Bi2201), $Bi_2Sr_2CaCu_2O_8$ (Bi2212), and $La_2CuO_4$(LCO) at three key points in the Brillouin zone ($\Gamma$, $X$ and $M$), and present the full set of frequencies and modes. We discuss the modes relevant to superconductivity, as well as any soft modes that indicate a structural instability.

This paper is organized into the following sections. We first detail our process for calculation, including discussion of our pseudopotentials. Next, phonon calculation results are presented for materials LCO, Bi2201, and Bi2212, highlighting modes with links to superconductivity, as well as any observed soft modes. We discuss the implications of phonons coupling to structural deformations. In the appendix we provide comprehensive tables of calculated modes, including frequencies and plots of the atomic motions. Finally, we developed code

Table 1: Wyckoff positions in the $I4/mmm$ unit cell for $La_2CuO_4$, $Bi_2Sr_2CuO_6$, and $Bi_2Sr_2CaCu_2O_8$. [33]

| Atom | Coordinates | Wyckoff Position |
|------|-------------|------------------|
| | LCO ($a = 3.82$Å, $c = 13.22$Å) | |
| Cu | (0, 0, 0) | 2$a$ |
| O(1) | (1/2, 0, 0) | 4$c$ |
| La | (0, 0, 0.361) | 4$e$ |
| O(2) | (0, 0, 0.186) | 4$e$ |
| | Bi2201 ($a = 3.63$Å, $c = 24.88$Å) | |
| Cu | (0, 0, 0) | 2$a$ |
| O(1) | (1/2, 0, 0) | 4$c$ |
| Sr | (0, 0, 0.427) | 4$e$ |
| O(2) | (0, 0, 0.105) | 4$e$ |
| Bi | (0, 0, 0.187) | 4$e$ |
| O(3) | (0, 0, 0.315) | 4$e$ |
| | Bi2212 ($a = 3.82$Å, $c = 30.7$Å) | |
| Ca | (0, 0, 0) | 2$a$ |
| Cu | (0, 0, 0.951) | 4$e$ |
| O(1) | (1/2, 0, 0.051) | 8$g$ |
| Sr | (0, 0, 0.427) | 4$e$ |
| O(2) | (0, 0, 0.105) | 4$e$ |
| Bi | (0, 0, 0.187) | 4$e$ |
| O(3) | (0, 0, 0.315) | 4$e$ |

to translate the phonon modes from the Wigner-Seitz cell to the conventional cell, which is included as a supplement.[1]

## 2 Methods

### A Calculation details

The electronic structure and phonon modes were calculated using the DFT implementation in Quantum ESPRESSO (QE) [34, 35]. The compounds were evaluated in their $I4/mmm$ body-centered tetragonal structure. The atomic Wyckoff positions and lattice cell parameters are listed in Table 1. We used QE's implementation of space groups to represent the materials with a minimal set of atoms ensure only physical phonon modes were calculated and to minimize computation time. We used 110 Ry and 700 Ry for the wavefunction and charge density energy cutoffs, respectively, and used a $15 \times 15 \times 15$ k-point grid. These values produced structural energies which were self-consistent to at least 3 decimal places, which is the level of precision used to identify the relaxed lattice parameters.

An initial round of phonon calculations was conducted using PBEsol/USPP pseudopotentials [36], but this produced an unusually large number of soft phonon modes in Bi2201. This was remedied by using PBE/PAW pseudopotentials [37], which did not produce as many soft modes. The phonon modes in LCO did not display the same level of sensitivity.

For LCO, our initial structure derived lattice constants from Ref. [38] and atomic positions from Ref. [39]. The lattice constants were presented for an orthorhombic unit cell, whereas

---

[1]Software is available on GitHub at https://github.com/kemperlab/axsf-cell-conversion.git.

we use tetragonal for improved calculation performance. We used QE's structural relaxation program to obtain minimal-energy values at $a = 3.82$Å and $c = 13.2$Å, which are consistent with the parameters given in Refs. [38,39]. Likewise, the relaxed atomic positions remain quite close to the inputs, with the most substantial deviation being the in-plane oxygens, which we assign to a high-symmetry $z = 0$ position rather than $z = -0.007$ (in units of the $c$-axis lattice constant). We then performed an optimization of the unit cell parameters $a, c$, identifying an energy minimum for LCO at $a = 3.82$Å, and $c = 13.2$Å, similar to the parameters used by [40] and [41].

For Bi2201, we used values of $a = 3.63$Å and $c = 24.9$Å, which were derived from a more cursory structural calculation using QE's `vc-relax` process, starting with the values used by [42]. The resulting value for $a$ is a bit smaller than the original $a \simeq 3.81$Å, but the vertical atomic positions are quite similar, within .01 (units of $c$) at the worst [42].

For Bi2212, we used $a = 3.82$Å and $c = 30.7$Å from the Crystallography Open Database [43, 44] (adapted to a tetragonal setting), but only allowed atomic positions to relax. These remained near their experimental positions, again differing by at most .01$c$ [42]. The final atomic positions are given in Table 1.

After structural optimization, the phonon modes were obtained via the Quantum ESPRESSO's `PHonon` program, which perturbs the atoms slightly from their equilibrium positions to calculate interatomic force constants and thus the fundamental oscillation modes. For modes at the Γ point, the crystallographic acoustic sum rule was applied. All $a-b$ plane modes are doubly degenerate, leaving only $2N$ unique modes. To compare our frequencies to those available in the literature [40,41], we assigned our modes to those found by the other authors using their reported symmetry identification and calculated displacements when available.

## B   Unit Cell Conversion

The output of the DFT software yields phonon modes in the Wigner-Seitz cell of the crystal. For visualization of the phonon in the conventional unit cell, we convert the forces from the `axsf` file produced by Quantum ESPRESSO's `dynmat.x` program. Due to the degeneracy of all $a-b$ plane modes for these materials, we also choose to align forces along the unit cell axes. The revised `axsf` files are then plotted using XCrySDen [45]. The software we used for this is available as a supplement. The code parses the `axsf` output file for the atomic names, positions, and forces of each phonon mode. Optionally, it will overwrite these forces from the `dynmat.x` output file to correct for an issue with version 5.4 of Quantum ESPRESSO wherein forces pointing in opposite directions are all assigned the same sign in the `axsf` file, making it impossible to distinguish symmetric and antisymmetric modes. A sample input file containing the desired conventional unit cell is then parsed for atomic names and positions, and a `json` map file is used to assign forces to symmetrically identical atoms. Finally, forces are aligned to the $a$, $b$, and $c$ axes as follows.

In the first step, modes are identified as $c$-axis polarized or in-plane by finding the maximum force value among all atoms, and setting the mode as vertical if this force is in the $c$ direction or in-plane otherwise. During this process, atoms with larger vertical component forces than horizontal components have their horizontal components zeroed out, thus aligning the forces to the $c$-axis. In practice, these forces already have negligible in-plane components, this step serves to streamline later alignment. For modes determined to be horizontal, the forces are then aligned to the $a$ and $b$ axes. The code scans through all atoms in a cell to find one with nonzero in-plane forces. This atom's normalized force vector is defined as the first basis vector $\mathbf{b}_1$. The second basis vector is then defined as

$$\mathbf{b}_2 = \begin{bmatrix} 0 & -1 \\ 1 & 0 \end{bmatrix} \mathbf{b}_1 . \tag{1}$$

From these two basis vectors, a transformation matrix $\mathbf{T}$ is defined by

$$\mathbf{T} = \begin{bmatrix} \mathbf{b}_1 & \mathbf{b}_2 \end{bmatrix}. \tag{2}$$

The in-plane forces on all atoms in a cell are multiplied by $\mathbf{T}^{-1}$. Their new first and second coordinates are compared, and the magnitude of the force is assigned to whichever basis vector has the larger coordinate. The aligned forces are then multiplied by $\mathbf{T}$ to return to the original coordinate system. Finally, the newly aligned forces are set to lie along either the $a$ axis or $b$ axis based on which coordinate is larger.

Once the forces are aligned to the axes of the new unit cell, for phonon calculations at the $X$ or $M$ points the code will alternate inverting the forces for adjacent unit cells. At $X$, forces alternate along the $a$ direction only, while at $M$, the forces invert along both $a$ and $b$ directions.

## 3 Results

We present the calculated phonon frequencies for LCO, Bi2201, and Bi2212 in Tables A1-A3. Tables A1 and A2 also contain frequencies from the literature for comparison. In cuprate superconductors, superconductivity is thought to occur within the horizontal Cu-O(1) planes. Thus, for each material, we describe in detail the atomic displacements for the phonon modes which influence the formation of a superconducting state at the Γ point. In addition, for LCO and Bi2201, we present plots of these modes at the X and M points of the Brillouin Zone. These modes are shown in Figs. 1-3. We follow with a brief discussion of particular modes of interest. The remaining modes are shown in Appendix A.

## A Phonon modes involving the $CuO_6$ octahedra

### A.1 Bond-buckling modes

The bond-buckling mode involves motion of the O(1) atoms along the $c$-axis in an alternating fashion. It is referenced in the literature for ties to kinks in the electron dispersion [19, 20], as well as a potential role in the emergence of a CDW order [31]. It is the only $B$ symmetry mode for the materials under study. The atomic displacements as well as the frequencies are shown in Fig. 1. For LCO, frequencies clustered tightly around 180 cm$^{-1}$, indicating a mostly non-dispersive optical mode. For Bi2212, the Γ-point modes had frequencies near 125 cm$^{-1}$. The buckling mode is soft for Bi2201, with frequencies near $50i$ cm$^{-1}$, which will be further discussed in Sec. B.2.

### A.2 Bond-stretching modes

The bond-stretching mode involves the O(1) atoms moving in the horizontal plane against the Cu atom, altering the Cu-O bond length. The atomic displacements as well as the frequencies are shown in Fig. 2. LCO had frequencies at all points close to 668 cm$^{-1}$, while frequencies for Bi2201 were near 810 cm$^{-1}$. The odd ($u$) and even ($g$) symmetry versions this mode in Bi2212 had frequencies of 694 cm$^{-1}$ and 624 cm$^{-1}$, respectively, indicating a modest coupling between the two planes through this mode. For all compounds, this was the highest energy mode, and for that reason, Le Tacon et al. [30] noted that it should contribute less to electron-phonon coupling than lower energy modes.

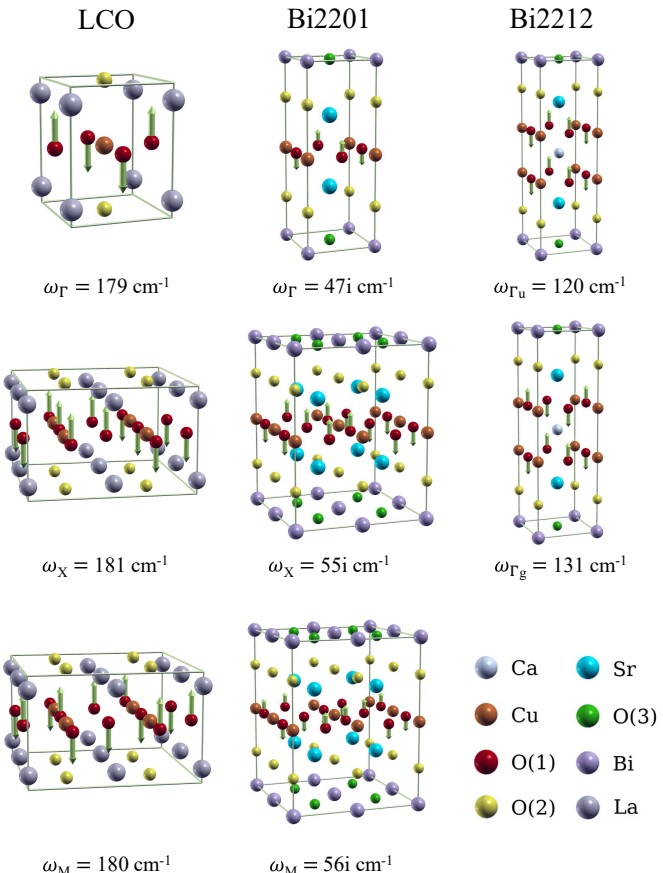

Figure 1: Cu-O bond buckling modes for $La_2CuO_4$(LCO), $Bi_2Sr_2CuO_6$(Bi2201) and $Bi_2Sr_2CaCu_2O_8$(Bi2212) at the $\Gamma$, $X$ and $M$ points in the Brillouin zone ($\Gamma$-only for Bi2212). Note that the buckling modes for Bi2201 frequencies are imaginary at all three points, indicating a structural instability across the Brillouin zone.

## A.3 Apical oxygen modes

This mode involves the O(2) atoms moving in opposite directions along the c-axis, creating an elongation/compression in the Cu-O octahedra. This high frequency mode was more variable across different BZ points than the other modes. It has been suggested [23] as being potentially involved with the 50-80 meV electron dispersion kink alongside the bond stretching mode, though the stretching mode likely dominates the effect. The mode is plotted in Fig. 3 for each material. For LCO, the frequencies were centered around 400 $\text{cm}^{-1}$. Bi2201 had less variability, with the mode having frequencies near 540 $\text{cm}^{-1}$ at all three BZ points, while in Bi2212, the mode had a frequency of 563 $\text{cm}^{-1}$.

## B Soft Modes

Soft modes, i.e. modes that result in an imaginary frequency, are associated with a structural instability. Since the calculations here have been performed in a high temperature tetragonal structure, we expect the soft modes to be indicative of the lower symmetry structures that appear for the compounds.

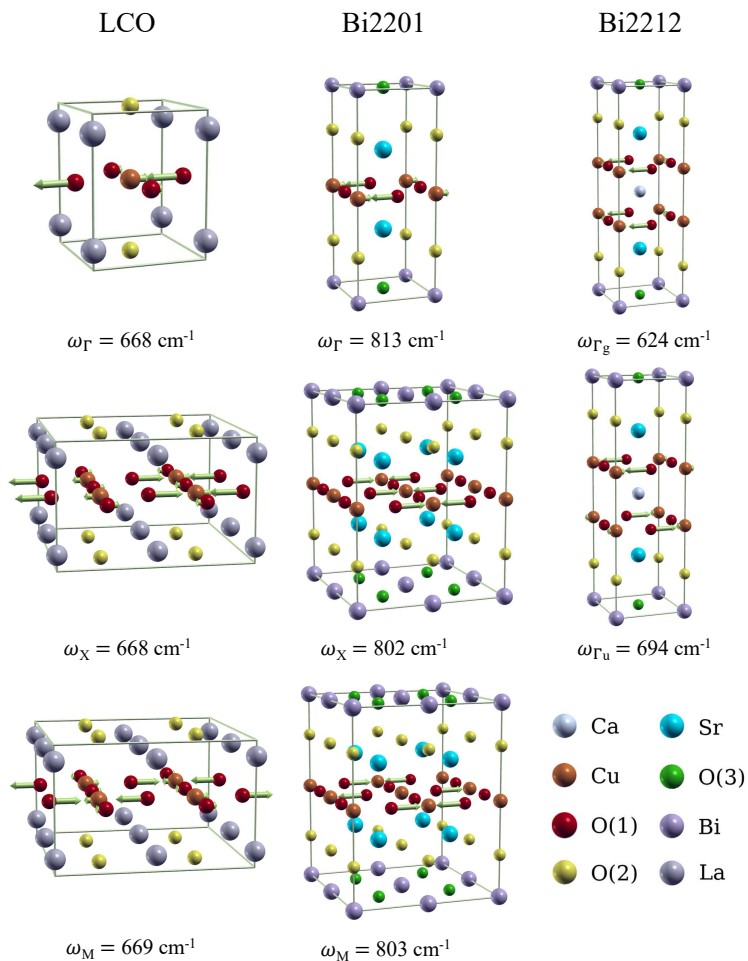

Figure 2: Cu-O bond stretching modes for $La_2CuO_4$(LCO), $Bi_2Sr_2CuO_6$(Bi2201) and $Bi_2Sr_2CaCu_2O_8$(Bi2212) at the $\Gamma$, $X$ and $M$ points in the Brillouin zone ($\Gamma$-only for Bi2212).

## B.1 Soft modes for $La_2CuO_4$

We identified one mode with an imaginary frequency across $\Gamma$, $M$, and $X$ points: an $E_g$ mode involving horizontal translations of the apical O(2) atoms. Singh et al. [41] also found this mode to be soft, with $\omega = 0$ residing within the margin of error for their calculated frequency, and identified an imaginary frequency for the $E_u$ mode involving the same atoms. As described in Ref. [46], LCO is not tetragonal at low temperatures and has tilted $CuO_6$ octahedra. Ref. [41] suggests that the mode in question displays instability due to strong coupling of phonons to this structural transition. This explanation is consistent with present results, as the soft mode involves a tilting motion by the apical oxygen atoms.

At X and M, we identified one additional soft mode, which was assigned to the horizontal acoustic mode. This further supports our expectation that the structure would undergo a structural phase transition involving in-plane deformations.

## B.2 Soft Modes for $Bi_2Sr_2CuO_6$

Bi2201 also exhibits a number of soft modes. Two modes involving $c$-axis motion of O(3) appear soft at $\Gamma$, while two other modes which appear soft only at X and M were assigned to

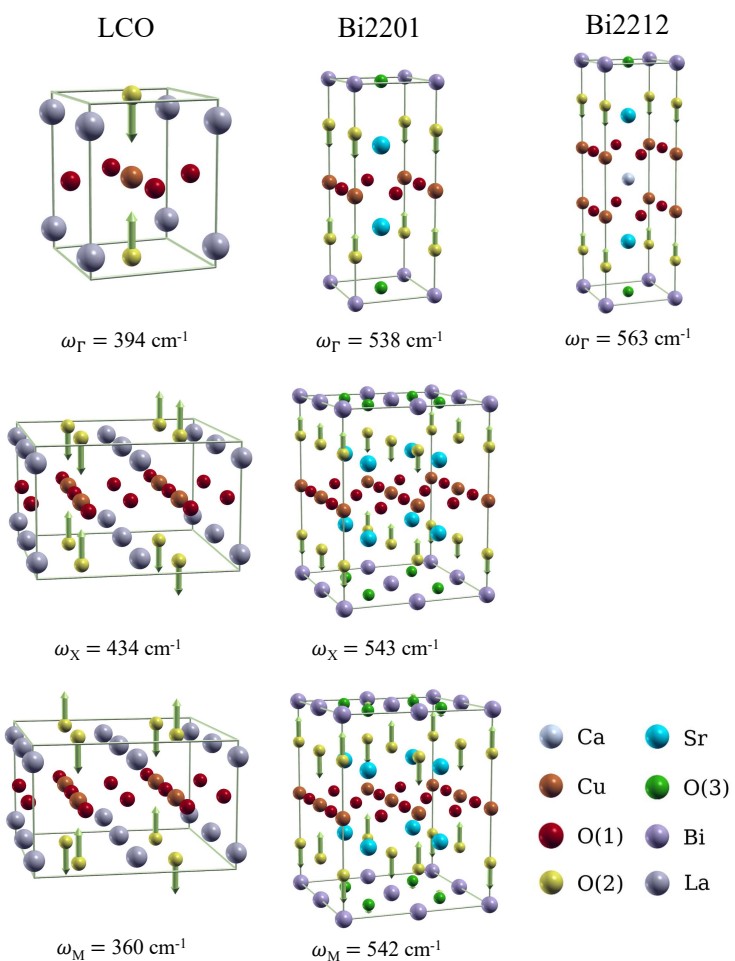

Figure 3: Cu-O apical oxygen modes for $La_2CuO_4$(LCO), $Bi_2Sr_2CuO_6$(Bi2201) and $Bi_2Sr_2CaCu_2O_8$(Bi2212) at the $\Gamma$, $X$ and $M$ points in the Brillouin zone ($\Gamma$-only for Bi2212).

the acoustic modes. In addition to the bond-buckling mode referenced previously, two modes were soft at all three BZ points, these involving horizontal motion of the O(3) atoms. Bi2201 also does not adhere perfectly to a tetragonal structure; as discussed in Ref. [47] Bi-O(3) bond lengths differ throughout a modulated superlattice, supporting the notion that these soft modes could result from coupling to structural instabilities.

### B.3 Soft Modes for $Bi_2Sr_2CaCu_2O_8$

Bi2212 exhibited more soft modes than the other two compounds. Some of these are similar to the soft modes seen in LCO and Bi2201, including horizontal motions of the O(2) and O(3) atoms, while others involve the Ca and Sr atoms, as well as the Cu-O plane. It is likely that coupling to known structural modulations [48] induces the softness of these modes.

# 4   Conclusion

Phonon modes were calculated for three high-temperature superconductor cuprates at multiple important Brillouin Zone points using DFT. Frequencies and symmetry assignments for all modes were presented, as well as plots of the force vectors for each mode. Good agreement with past work [40,41] was obtained for LCO phonon frequencies, but frequencies for Bi2201 are considerably lower than those reported by Ref. [42]. This index of modes is useful for work attempting to identify the effects of a specific phonon mode near a given energy. Multiple soft modes were obtained which likely stemmed from coupling between phonons and structural instabilities, indicating that the materials have complicated modulated structures at low temperatures. Our description of the calculation process may be useful for investigating the phonons of other materials.

# Acknowledgements

Our research used computational resources from the National Energy Research Scientific Computing Center (NERSC), a U.S. Department of Energy Office of Science User Facility operated under Contract No. DE-AC02-05CH11231.

**Funding Information**   A.F.K. and N.J.J. were supported by the National Science Foundation under Grant No. DMR-1752713 and by the NC State University Provost's Professional Experience Program.

# A   Complete mode tables

For completeness, we present the full set of atomic displacements and corresponding oscillation frequencies for all three compounds.

Table A1: LCO mode frequencies and assignments at $\Gamma$, $X$ and $M$. All mode frequencies are in cm$^{-1}$.

| | | | | | | | | |
|---|---|---|---|---|---|---|---|---|
| LCO | | | | | | | | |
| $\Gamma$ | | | | | X | | M | |
| Assignment | $\omega$ | $\omega$ [41] | $\omega^2$ | $\omega^3$ | Assignment | $\omega$ | Assignment | $\omega$ |
| $E_g$ | $62i$ | 15 | 26 | 91/90 | $E_g$ | $34i$ | $E_u$ | $93i$ |
| $E_u$ | 29 | $75i$ | 22 | 126/160 | $E_u$ | $29i$ | $E_g$ | $59i$ |
| $A_{2u}$ | 127 | 119 | 132 | 149/150 | $E_g$ | 77 | $A_{2u}$ | 26 |
| $E_u$ | 159 | 146 | 147 | 173/177 | $E_u$ | 92 | $E_u$ | 28 |
| $E_g$ | 165 | 212 | 201 | 241/243 | $A_{2u}$ | 109 | $A_{2u}$ | 110 |
| $B_{2u}$ | 179 | 201 | 193 | 270/264 | $A_{1g}$ | 126 | $E_u$ | 159 |
| $A_{2u}$ | 194 | 197 | 182 | 251/410/310 | $A_{2u}$ | 170 | $E_g$ | 168 |
| $A_{1g}$ | 218 | 215 | 202 | 227 | $B_{2u}$ | 181 | $B_{2u}$ | 180 |
| $E_u$ | 332 | 312 | 319 | 354/347 | $E_u$ | 218 | $A_{2u}$ | 203 |
| $A_{1g}$ | 394 | 390 | 375 | 427 | $E_u$ | 317 | $A_{1g}$ | 203 |
| $A_{2u}$ | 458 | 446 | 441 | 497 | $A_{2u}$ | 362 | $E_u$ | 318 |
| $E_u$ | 668 | 650 | 630 | 684/680 | $A_{2u}$ | 426 | $A_{1g}$ | 360 |
| | | | | | $A_{1g}$ | 434 | $A_{2u}$ | 458 |
| | | | | | $E_u$ | 668 | $E_u$ | 669 |

---

$^2$ [40] DFT data.
$^3$ [40] Neutron scattering data.

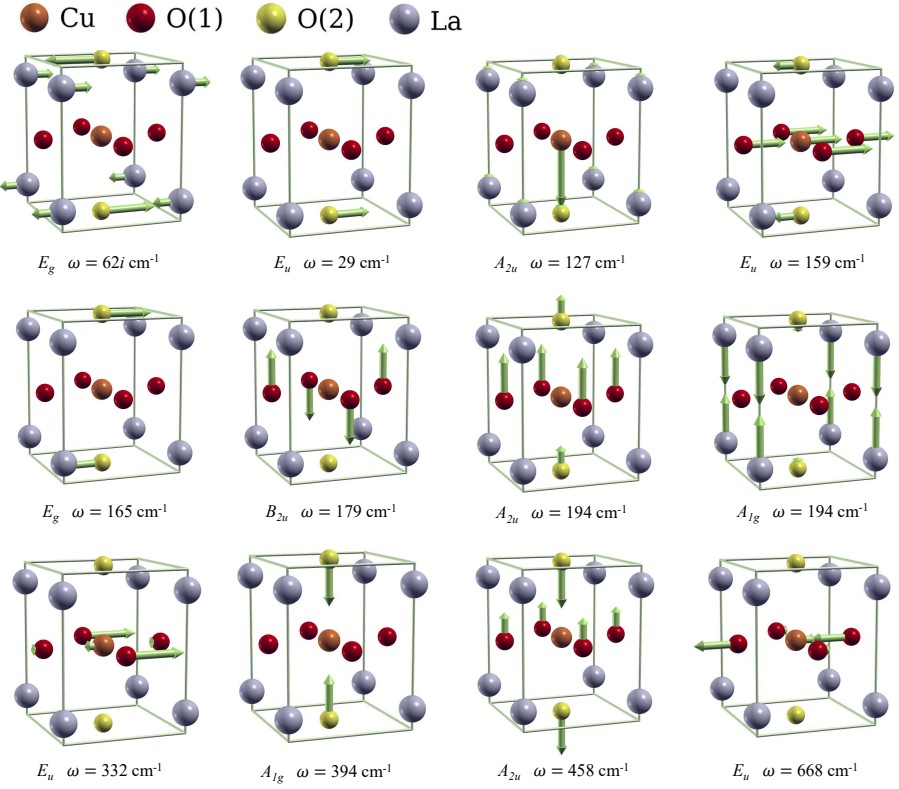

Figure A1: All optical LCO modes at the Γ point.

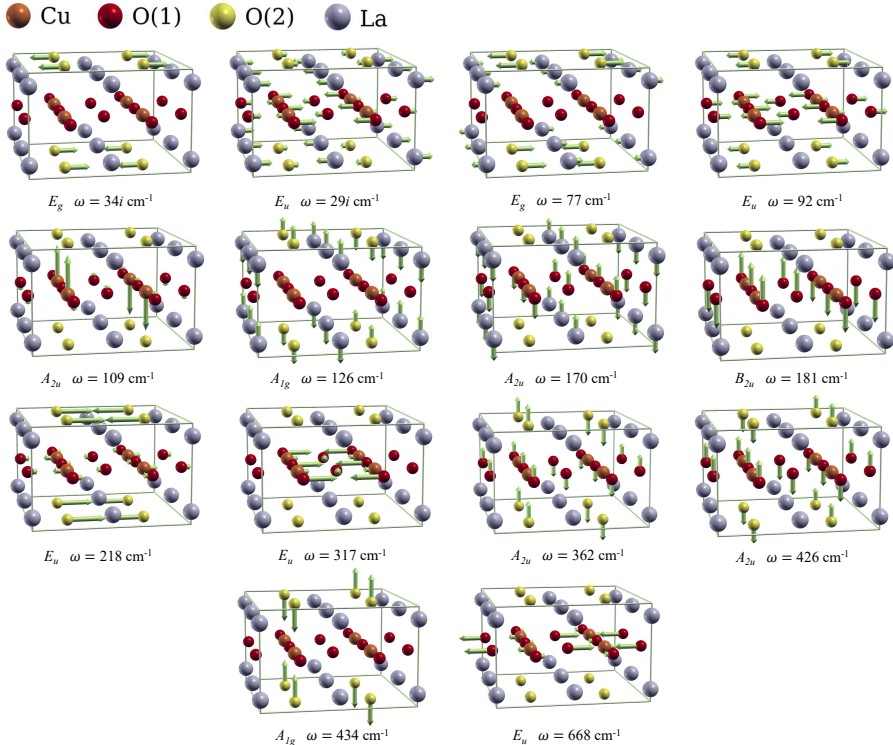

Figure A2: All LCO modes at the $X$ point, represented in a 2x2 supercell.

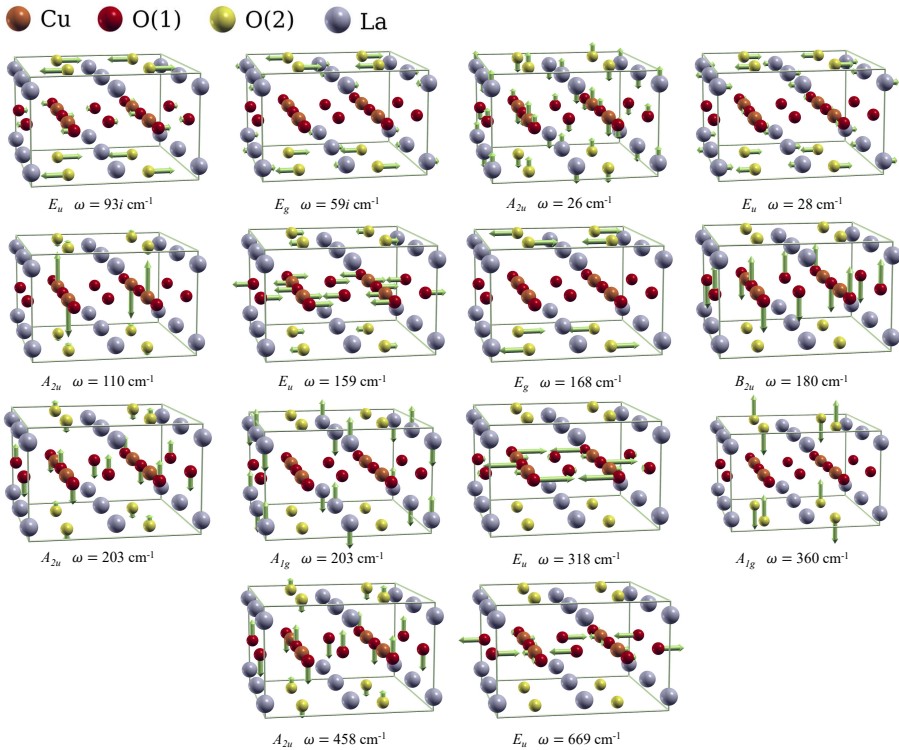

Figure A3: All LCO modes at the $M$ point, represented in a 2x2 supercell.

Table A2: Bi2201 mode frequencies and assignments at $\Gamma$, $X$ and $M$. All mode frequencies are in cm$^{-1}$.

| Bi2201 | | | | | | |
|---|---|---|---|---|---|---|
| $\Gamma$ | | | X | | M | |
| Assignment | $\omega$ | $\omega$ [42] | Assignment | $\omega$ | Assignment | $\omega$ |
| $E_g$ | 204i | | $E_u$ | 201i | $E_g$ | 202i |
| $E_u$ | 175i | | $E_g$ | 172i | $E_u$ | 172i |
| $A_{2u}$ | 138i | 175 | $E_u$ | 92i | $A_{2u}$ | 107i |
| $A_{1g}$ | 78i | 111 | $B_{2u}$ | 55i | $E_u$ | 71i |
| $B_{2u}$ | 47i | | $A_{2u}$ | 28i | $B_{2u}$ | 56i |
| $E_g$ | 21 | | $E_g$ | 51 | $E_g$ | 10 |
| $E_u$ | 35 | | $A_{1g}$ | 66 | $E_u$ | 60 |
| $E_g$ | 85 | | $E_u$ | 70 | $E_g$ | 81 |
| $A_{2u}$ | 109 | 77 | $E_g$ | 88 | $A_{2u}$ | 90 |
| $E_u$ | 124 | | $A_{2u}$ | 94 | $A_{1g}$ | 99 |
| $A_{2u}$ | 162 | 293 | $A_{2u}$ | 131 | $A_{1g}$ | 143 |
| $A_{1g}$ | 187 | 229 | $E_u$ | 138 | $E_u$ | 151 |
| $E_g$ | 198 | | $A_{1g}$ | 178 | $A_{2u}$ | 159 |
| $E_u$ | 227 | | $E_u$ | 197 | $A_{2u}$ | 190 |
| $A_{1g}$ | 231 | 423 | $E_g$ | 208 | $E_g$ | 194 |
| $A_{2u}$ | 237 | 353 | $A_{2u}$ | 211 | $E_u$ | 203 |
| $E_u$ | 282 | | $A_{2u}$ | 267 | $A_{1g}$ | 254 |
| $A_{1g}$ | 538 | 575 | $A_{1g}$ | 271 | $E_u$ | 283 |
| $A_{2u}$ | 545 | 596 | $E_u$ | 281 | $A_{2u}$ | 295 |
| $E_u$ | 813 | | $A_{2u}$ | 541 | $A_{2u}$ | 526 |
| | | | $A_{1g}$ | 543 | $A_{1g}$ | 542 |
| | | | $E_u$ | 802 | $E_u$ | 803 |

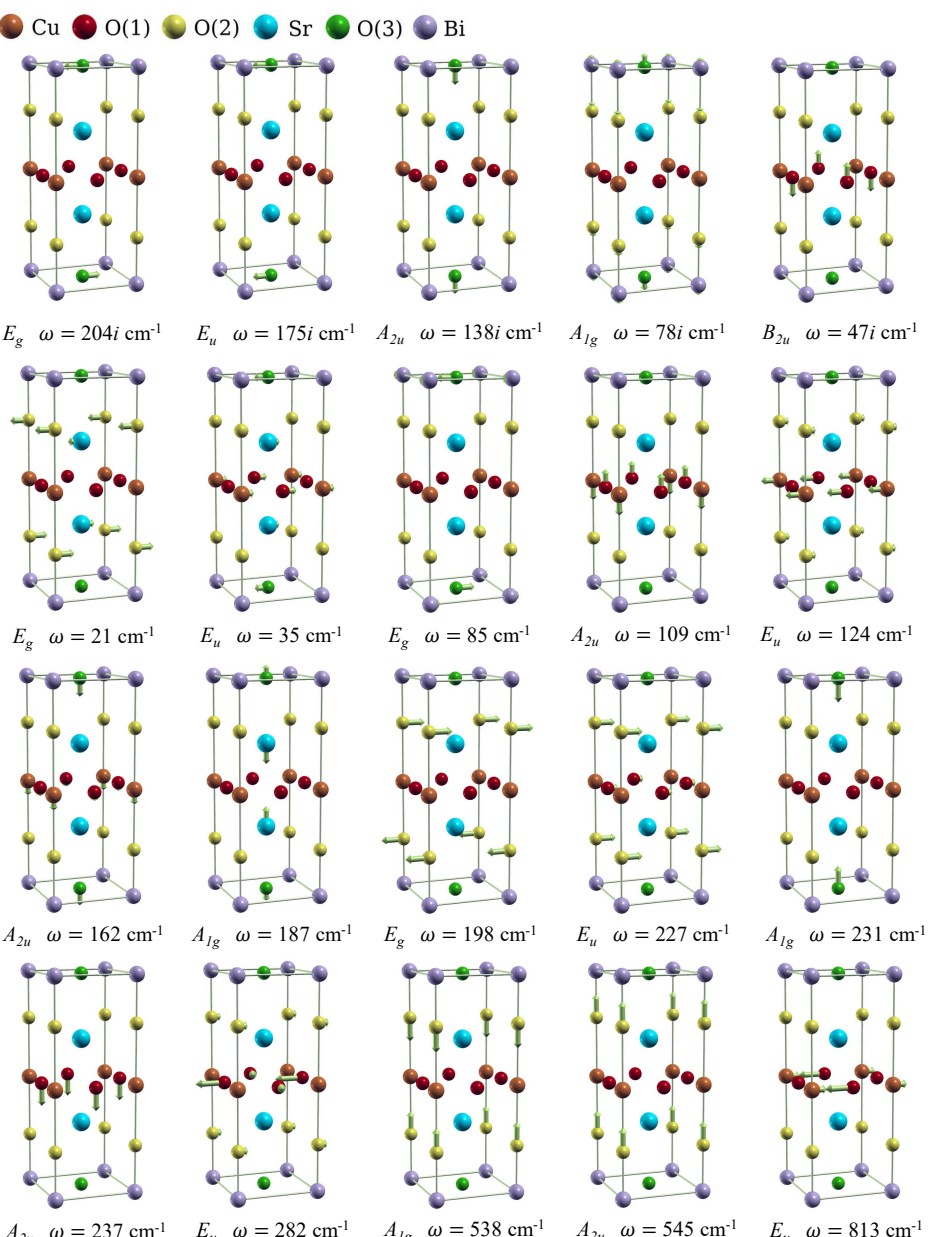

Figure A4: All optical Bi2201 modes at the Γ point.

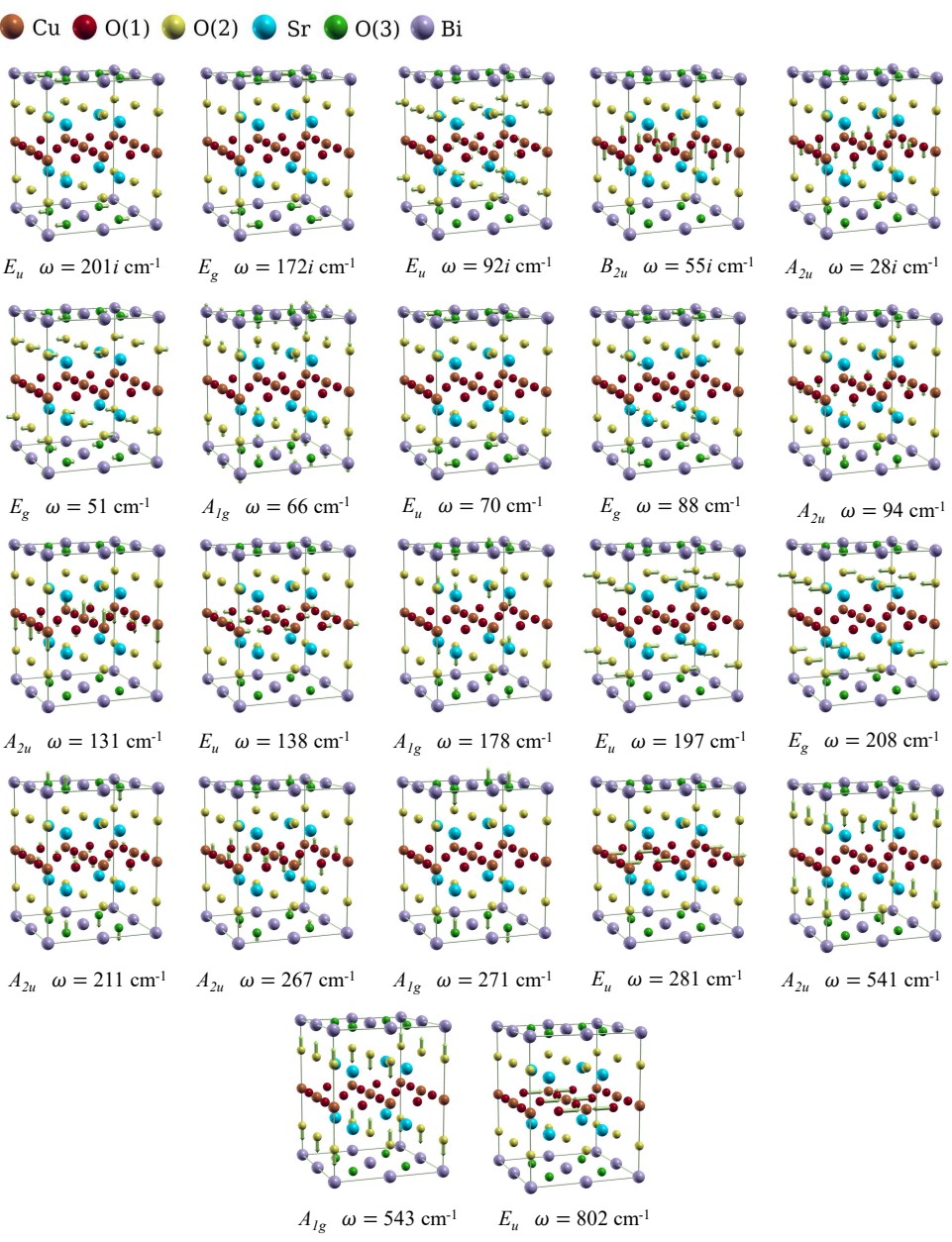

Figure A5: All Bi2201 modes at the $X$ point.

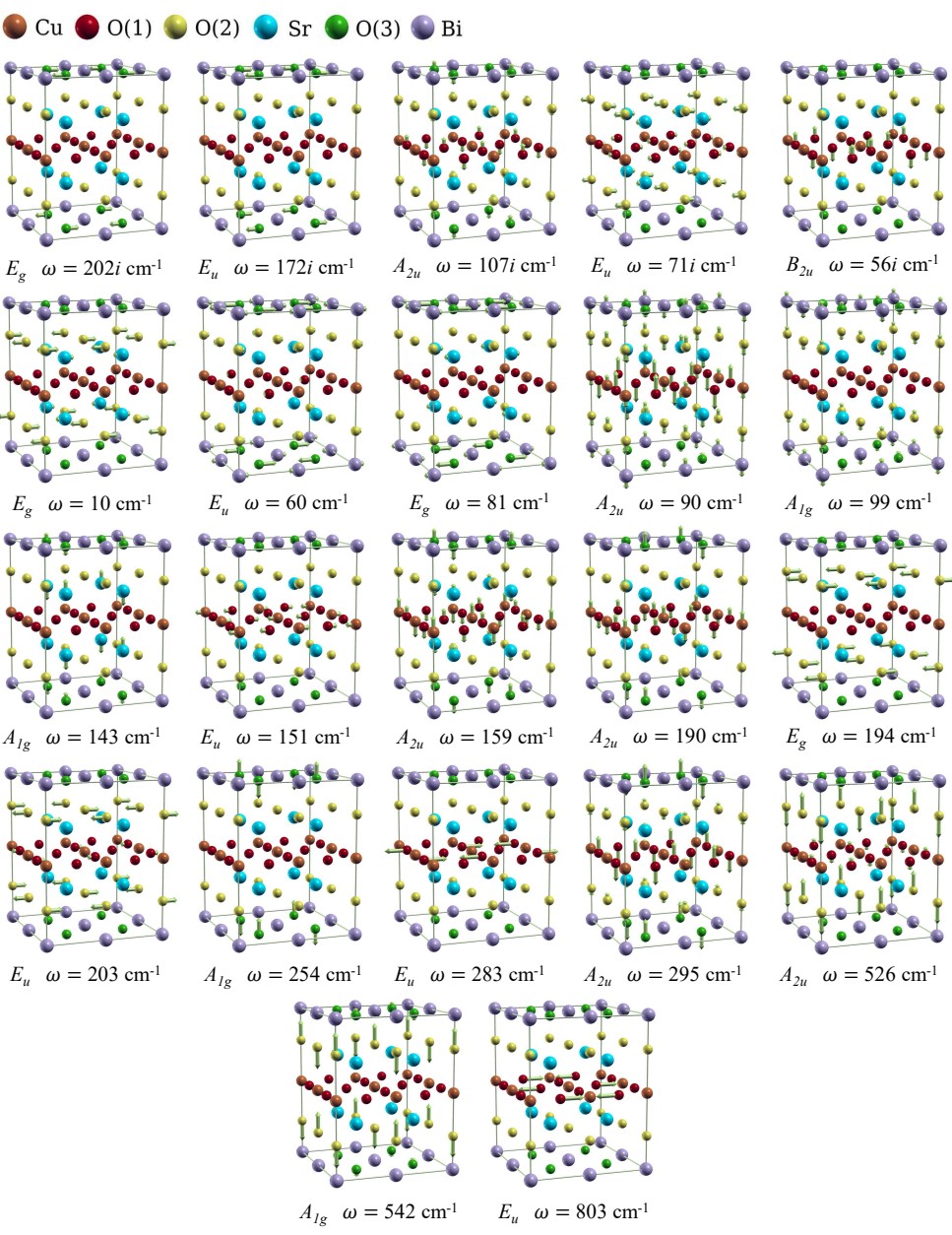

Figure A6: All Bi2201 modes at the $M$ point.

Table A3: Bi2212 optical mode frequencies and assignments at Γ. All mode frequencies are in cm$^{-1}$.

| Bi2212 | |
|:---:|:---:|
| Assignment | $\omega$ |
| $E_u$ | 394$i$ |
| $E_g$ | 247$i$ |
| $E_u$ | 204$i$ |
| $E_u$ | 126$i$ |
| $E_g$ | 107$i$ |
| $E_u$ | 60$i$ |
| $E_g$ | 54$i$ |
| $A_{2u}$ | 57 |
| $A_{1g}$ | 77 |
| $E_g$ | 85 |
| $B_{2u}$ | 120 |
| $B_{1g}$ | 131 |
| $A_{1g}$ | 136 |
| $A_{2u}$ | 149 |
| $E_g$ | 150 |
| $A_{1g}$ | 159 |
| $E_u$ | 170 |
| $E_u$ | 200 |
| $A_{1g}$ | 223 |
| $A_{2u}$ | 224 |
| $A_{2u}$ | 259 |
| $A_{2u}$ | 282 |
| $E_g$ | 308 |
| $A_{1g}$ | 324 |
| $A_{2u}$ | 553 |
| $A_{1g}$ | 563 |
| $E_g$ | 624 |
| $E_u$ | 694 |



Figure A7: All optical Bi2212 modes at the Γ point.

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
