# Peer review of "Catalogue of phonon modes in several cuprate high-temperature superconductors from density functional theory"

_SciPost Physics Core, doi:SciPost Phys. Core 6, 018 (2023)_

## Round 1 · Referee Report · Anonymous (Referee 1) · 2022-12-1

Report

The authors made a significant improvement of the manuscript following the referee's recommendation. I think readability of the text has improved and I have no further reservation to recommend the publication.

---

## Round 1 · Author Response

Dear editor,

We appreciate the feedback received and the opportunity to resolve the indicated weaknesses. We have clarified the details of our determination of the crystal structure, adding a direct comparison between the values of lattice constants and atomic positions used with those reported in the literature.

In regards to the reviewer’s second point, due to time constraints we are unable to complete magnetic calculations or those required for generating plots for the density of states F(\omega). We indeed found negative/imaginary frequency modes, though these are correlated to the materials’ known structural phase transitions at low temperature (we did our calculations in the tetragonal phase). We have added references to the Bohnen paper to situate our own work in relation to the suggested one.

Finally, we have added substantial discussion and several background sources about the role of electron-phonon interactions in cuprates generally and their impact on superconductivity specifically, as requested by the third review concern.

Below, we provide a point-by-point response to the referee.

Sincerely,

N. Jabusch (for the authors)

---

## Round 1 · List of Changes

Received Report:
Strengths
1) the paper presents a catalogue of the phonon modes in three representative high-Tc cuprate superconductor based on the DFT.
2) the calculations are strightforward
3) the presentation is clear
Weaknesses
1) it is not clear how close the relaxed structure used in these calculations is to the actual experimentally measured (especially for the oxygen position). The problem is that even a few percent differences might become crucail for the phonons. Given that 214 is an antiferromagnetic insulator whereas it is a non-magnetic metal in the DFT I am a bit concerned about the accuracy of the calculations. In any case the authors should discuss this.

When performing density functional theory calculations, it is indeed often the case that both the lattice constants and internal degrees of freedom do not precisely match the experimental results. In essence, this is due to the inexact description of DFT. The typical approach is to ensure that at least the DFT result is stable – this is achieved by minimizing both the lattice parameters and atomic positions. In our work, we performed one or both of these steps (depending on computational time availability), and find results that are in reasonable agreement with the initial (experimental) inputs.
We have added further details that address these points in the calculation details in Sec. 2A.

2) could the authors also show not only the tabular frequencies of various phonons but also the spectrum plot F(\omega) to see if the phonons are overall stable (i.e. there are no negative dispersions) and discuss this in the manuscript. I am a bit afraid that non-
magnetic DFT calculations can show some negative dispersions, signalling some problems in not accounting for the effect of the magnetism in the calculations. For the reference the authors could orient themselves to the classical work
Europhys. Lett., 64 (1), pp. 104-110 (2003)

The referee is correct that negative frequencies can appear due to either a neglect of magnetism in the calculations, or due to structural instabilities. In fact, we do find negative frequencies at the high symmetry points in the Brillouin zone that correspond, which we term “soft modes.” They are discussed in detail in Sec. 3B.
Due to time constraints, we are limited to the calculation of the phonon modes at the high-symmetry points, and cannot investigate the spectrum/dispersion.

3) Overall I think the Introduction will benefit from having
a bit broader discussion on the role of phonons in the field of High-Tc and the ambiguity of the alpha^2F(omega) extracted from DFT and claimed in the recent experiments. In the context of YBaCuO superconductors some works on the phonons have been made

We appreciate the referee’s suggestion. We have incorporated a discussion of these issues in the introduction (section 1), and added additional references concerning the study of YBCO.

Report
Overall I believe the paper might be interesting provided the authors make an iteration to improve the presentation reflecting the weak points (critical remarks) outlined above. Once the authors provide a satisfactory explanation the paper can be recommended for publication

We thank the referee for the positive assessment.

---

## Editorial Decision

published